# Low-Dimensional Embedding Techniques for Knowledge Graph Mapping and LLMs

Liubov Tupikina[,†]

[1] *Nokia Bell labs, France,* [2] *University Paris Descartes, Paris, LPI, France,*

## Abstract

We propose a framework, which aims at construction of explainable data embedding methods. It is specifically based on the low-dimensional embedding techniques which are connecting higher-order geometric analysis, topological data analysis and natural language processing methods.
We consider the applications of our framework to the navigation of the knowledge scape is non-trivial in the everyday context, when knowledge/data growth is beyond exponential. Moreover our framework supports methods for generating knowledge graph (node) embeddings, and temporal knowledge graph embeddings.

**Keywords** LLMs New Approaches for Combining Deep Learning, LLMs, and Knowledge Graphs

## 1. Introduction

In the era of information abundance, navigating the knowledge scape and utilizing large language models (LLMs) trained on vast amounts of textual data has become a significant challenge [1-3]. The exponential growth of data sources necessitates the development of robust and explainable dimensionality reduction techniques to process this high-dimensional data effectively.

Despite their remarkable performance in various natural language processing tasks, large language models (LLMs) [3, 6] are trained on general-purpose data and have lower performance in domain-specific tasks. Additionally, LLMs suffer from hallucination problems, and they are opaque models that lack interpretability. A potential solution to these problems is to induce the knowledge from knowledge graphs to LLMs [6]. TDA studies the shape by going beyond the standard measures defined on data points pairs. We move from networks with simple edges to simplicial complexes. Simplicial complexes are obtained from elementary objects called simplices. Simplices can be built from simple points, line segments, triangles, and other higher order structures all glued together. This representation in latent space provides more interpretable results to visualize relationships in data. To uncover hidden patterns and relationships in data that may not be apparent through traditional analysis methods, we explore higher-order

_______________________________

[1*] Corresponding author. liubov.tupikina@nokia-bell-labs.com
ORCID number 0000-0002-7169-5706

data analysis [8]. By considering higher-order interactions, we aim to gain deeper insights into complex datasets.

This paper proposes the use of low-dimensional embedding techniques for working with LLMs at various stages, including training and analysis of latent space. These techniques are crucial for understanding and managing the complex relationships between entities in the knowledge graph, which is essential for effective knowledge representation and reasoning [6].

The need for such techniques arises from the fact that LLMs are trained on massive datasets, making it difficult to interpret their behavior and decisions. By reducing the dimensionality of the data, we can gain insights into the underlying patterns and structures that govern the behavior of these models. In the contemporary era, where knowledge and data growth is beyond exponential, navigating the knowledge scape becomes a non-trivial task. To address this challenge, in this methodological survey paper we propose to use low-dimensional embedding techniques connected to knowledge graph mapping and grounded in the theory of topological data analysis and higher-order structures. This approach aims to provide a more comprehensive understanding of complex relationships within high-dimensional datasets and specifically in the context of designing explainable frameworks for LLMs. In the Methods **section** we explain about the main methodological **framework of topology infused embeddings.** In the Results and Discussions **section** we describe the main methodological advances and possible perspectives.

**Section Methods**

Our proposed approach for explainable mathematical foundations for AI systems models, such as LLMs, is based on the idea of combination of neural network latent space investigation using hypergraph approaches and dimensionality reduction techniques. For demonstrating our approach we use several embedding models (BERT, general autoencoder architectures) [3] applying them to the high-dimensional textual data with some ground-truth information about the datasets. By ground-truth we mean the information about the interconnection in data. **The main idea behind** the method is that then using our approach the results of the embeddings to texts then can be studied as geometrical structures, such as manifolds or hypergraphs (***Figure 1***), which generally simplifies and creates yet another language for analysis of LLMs and evaluation of LLMs.

The method of topology infused embeddings consists of several main stages:

- Data preprocessing: we preprocess the data by cleaning and normalizing it to improve the quality of the input.

- Embedding models, when we use embedding models, such as BERT or general autoencoder architectures, to transform the high-dimensional textual data into low-dimensional embeddings. These embeddings capture the semantic relationships between words and phrases in the text.

- Next, we study the neural network latent space using higher-order data analysis (hypergraph theory) and geometry infused topological properties of such structures.
  We analyze the embeddings using topological methods, such as manifold learning or hypergraph analysis. These methods allow us to identify patterns and structures in the data that are not visible in the original high-dimensional space. Then we interpret the results of the topological analysis to understand the underlying semantics of the data. For example, we can identify clusters of similar words or phrases, or we can visualize the relationships between different concepts.

The topological properties of some of the regions of such embeddings then can be directly linked and related to the projected content from texts, which helps us to relate the higher-order information (in this context, it is text) with some geometrical and topological properties of such learned embedded objects (such as manifolds). Overall we believe that this brings us to the next level of data analysis, where neural networks architectures of embedding methods can be linked with topological data analysis dimensionality reduction techniques [4,5]. Hence this may be a pathway towards resolving some aspects of explainability issues in LLMs or over-reliance on training data in LLMs, since with such an approach we aim to find some invariant correspondence between textual and non-textual structures. The system may not have been trained directly on the benchmark, but on similar items that necessitate comparable reasoning patterns. We extend our method by investigating some of the benchmarks and knowledge databases. Moreover we demonstrate its applications in mapping of news ideas that appear through time and how knowledge grows [1], as well as generally map and represent high dimensional (textual) data in the lower dimensional representation with the explainable AI methods. In one of the previous applications of our method in previous research papers [1] we looked at millions of articles in science, to see how the knowledge landscape evolves [1,2].

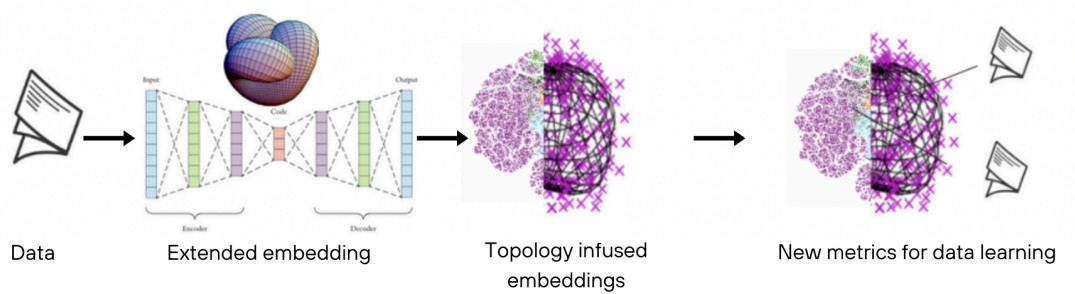

Data          Extended embedding      Topology infused      New metrics for data learning
                                               embeddings

*Figure 1.* The development of the method, where the data used for training LLMs is curated using low-dimensional topology infused embedding methods. At the later stages of the LLMs generating content, this would also enable analysis of the latent space of LLMs (here we consider specifically the autoencoder type of architecture).

**Section Discussions and conclusions**

One of the key advantages of higher-order interaction analysis is its ability to capture non-linear and non-monotonic relationships between variables. This allows us to model more complex phenomena and make more accurate predictions. For example, higher-order interactions could reveal how the combination of different demographic factors (e.g., age, gender, income) influences consumer behavior, or in the context of scientometrics project [1,2] it can be simply a combination of new scientific fields emerging.

The approach combines neural network latent space investigation using hypergraph approaches and dimensionality reduction techniques, which allows us to study the results of embeddings as geometrical structures. This simplifies and creates yet another language for analysis of LLMs, making it easier to relate higher-order information (text) with geometrical and topological properties of embedded objects.

This method can be used to explore the potential for using LLMs as tools for knowledge discovery and information retrieval. By mapping news ideas that appear over time, we can see how knowledge grows and evolves. This approach can also be applied to other types of high-dimensional data, such as scientific articles or social media posts.

In conclusion, the proposed method is a promising approach for explainable mathematical foundations for AI systems models, such as LLMs. It provides a framework for understanding how LLMs process and generate text, and it can be used for exploring the potential of LLMs for knowledge discovery and information retrieval.

Future research could focus on applying this method to more diverse datasets and exploring its limitations and potential applications.

**Citations and bibliographies**

[1] C. Singh, L. Tupikina, M. Starnini, M. Santolini "Charting mobility patterns in the scientific knowledge landscape" Nat.Comms arxiv.org/abs/2302.13054 arxiv, EPJ (2024)

[2] C. Singh, E. Barme, R. Ward, L. Tupikina, M. Santolini "Quantifying the rise and fall of scientific fields", Plos One 17(6): (2022)

[3] Maarten Grootendorst. BERTopic: Neural topic modeling with a class-based TF-IDF procedure. (arXiv:2203.05794). URL http://arxiv.org/abs/2203.05794

[4] Dmitry Kobak and Philipp Berens. The art of using t-SNE for single-cell transcriptomics. 10(1):5416. ISSN 2041-1723. doi: 10.1038/s41467-019-13056-x. URL https://www.nature.com/articles/ s41467-019-13056-x.

[5] Patania, A., Vaccarino, F. & Petri, G. Topological analysis of data. EPJ Data Sci. 6, 7 (2017). https://doi.org/10.1140/epjds/s13688-017-0104-x

[6] M Alam, F van Harmelen, M Acosta    Towards Semantically Enriched Embeddings for Knowledge Graph Completion arXiv preprint arXiv:2308.00081 (2023)

[7] McInnes, Leland & Healy, John. (2018). UMAP: Uniform Manifold Approximation and Projection for Dimension Reduction.

[8] HyperNetX documentation Basic 5 - HNX attributed hypergraph.ipynb - Colab (google.com)
