# OpenReview forum: "Dimensional Embedding Techniques for Knowledge Graph Mapping and LLMs"
_KDD.org/2024/Workshop/DL4KG — Submitted to DL4KG 2024_

### Official Review · Reviewer_uxGX · 2024-07-02
**Insuffiently precise to be of interest.**

**Rating:** 2
**Confidence:** 5

**Review:**

This paper appears to present a proposed research framework for the embedding of knowledge. There are no results or evaluations. A position paper on the development of such a framework, presented before the work is done, may be an interesting contribution to a workshop. However, in such a paper the following should at least be present:

1) A clear definition of the proposed framework, including clear definitions of of terms (such as "higher-order interactions", "knowledge scape" and "topology infused embeddings"). At the moment, the framework is not sufficiently precisely described for me to form any judgment.
2) An extensive embedding of the idea in the related work. In this case, there is a huge amount of literature on geometric learning using, for example, non-Euclidean embeddings, group theory or Clifford algebras, all of which may be relevant. The paper should clearly indicate these jumping off points and the novelty of the proposed approach.
3) Clear, falsifiable predictions. The paper should make clear predictions for what the proposed research program will achieve. These predictions should be stated clearly enough to be evaluated, and be precise enough to be falsifiable.

Since the paper fails to provide any of these, I can only recommend a strong reject. I expect that the idea has merit, and I hope the author will pursue this line of research, but it should be much further developed before being submitted for review.

---

### Official Review · Reviewer_PABT · 2024-07-02
**Interesting research direction but methodological details unclear**

**Rating:** 3
**Confidence:** 5

**Review:**

Dear author,

I find it very relevant to carry out research with the goal of making LLMs more understandable and combining LLMs with knowledge graphs. Reading your paper gives me the impression that you have plans to carry out research in this direction, but it seems like you have not yet worked out the details of the methods nor have you carried out experiments to validate your beliefs.

There a a couple of specific things that were not clear to me or that I'd like to mention:

1. "robust and explainable dimensionality reduction techniques". It seems like you regard textual data as being high-dimensional. Then, does dimensionality reduction mean that you train an LLM, for example to obtain vectors in, e.g., 300 dimensional space?

2. LLMs are trained on general-purpose data and have lower performance in domain-specific tasks. This is true, but these models can be fine-tunes for specific tasks.

3. You use the acronym "TDA" but you never explain what it is. Only later I understood that this probably is topological data analysis.

4. "This representation in latent space provides more interpretable results to visualize relationships in data". I do not understand what these representations are and what the role of simplicial complexes are. But aren't vector spaces in general difficult to understand?

5. The method is entirely unclear to me. You mention that you combine dimensionality reduction (which in your case means training an LLM) with knowledge graph mapping (I am unsure what that means. Is this about embedding entities and relations from a knowledge graph?), somehow making use of methods from topological data analysis (where it is not clear to me which methods, what they do, how they are combined). What really are topologically-infused embeddings? Is a KG relevant here? Figure 1 does not help me to understand the method.

6. I have the impression that the paper is not sufficiently self-contained. One should be able to understand the main idea without reading [1] and [2].

7. You did not use the official template, but I reviewed your paper anyways.

In general, as I wrote in the beginning, this is an interesting research direction. But it might be in a too early state for a publication. My recommendation is to carve out a research question or hypothesis for which data exists or can be created so that you can carry out an experiment to gain insights. Then develop the minimal core of your method and carry out an experiment. Maybe advance step by step. Maybe topology-infusion is already sufficiently interesting - no need for external knowledge from knowledge graphs (unless I misunderstand you and one needs external knowledge for topology infusion).

---

### Official Review · Reviewer_nJ67 · 2024-07-03
**Unstructured with no clear contribution.**

**Rating:** 1
**Confidence:** 3

**Review:**

The paper superficially describes the idea of using topological methods in analyzing and interpreting LLMs.

# Weak Points
- Technical details are missing. Scientific contribution is vague.
- Texts are repetitive.
- Methodology is lacking in detail.
- Proposed method is similar to (Park et al., 2024) https://arxiv.org/pdf/2406.01506
- The CEUR-WS LaTeX template was not used.

# Minor Comments:
- Page 1: What is TDA?
- Reference #8: Perhaps cite the paper instead of a collab notebook?

# Score
## Quality (1-5): 1 (very poor)
*Unstructured, lacking in literature review."

## Clarity (1-5): 1 (very poor)
*Proposed approach not clearly described or motivated."*

## Originality (1-5): 1 (marginal)

## Relevance to the Conference (1-5): 2 (only marginal relevance)
*The combination of graph theory and LLMs are relevant to this workshop. However, sans scientific contribution deems this relevance moot."

## Impact (1-5): 1 (very low)
*No clear contribution."

## Technical Soundness (1-5): 1 (very low)
*Lacking in technical details.*

---

### Decision · Program_Chairs · 2024-07-09

Reject